# LSTM Iteration Networks: An Exploration of Differentiable Path Finding

**Lisa Lee**[*] **& Emilio Parisotto**[*] **& Devendra Singh Chaplot & Ruslan Salakhutdinov**
Machine Learning Department
Carnegie Mellon University
Pittsburgh, PA 15213, USA
{lslee,eparisot,chaplot,rsalakhu}@cs.cmu.edu
[*] denotes equal contribution.

## Abstract

Our motivation is to scale value iteration to larger environments without a huge increase in computational demand, and fix the problems inherent to Value Iteration Networks (VIN) such as spatial invariance and unstable optimization. We show that VINs, and even extended VINs which improve some of their shortcomings, are empirically difficult to optimize, exhibiting instability during training and sensitivity to random seeds. Furthermore, we explore whether the inductive biases utilized in past differentiable path planning modules are even necessary, and demonstrate that the requirement that the architectures strictly resemble path-finding algorithms does not hold. We do this by designing a new path planning architecture called the LSTM-Iteration Network, which achieves better performance than VINs in metrics such as success rate, training stability, and sensitivity to random seeds.

## 1 Introduction

A common type of sub-task that arises in various reinforcement learning domains is path finding: finding a shortest set of actions to reach a subgoal from some starting state. Due to its ubiquity in important applications, recent work (Tamar et al., 2017) has designed a differentiable path-finding modules. These *Value-Iteration-Networks (VINs)* mimic the application of Value Iteration (VI) on a 2D grid world, but without pre-specified MDP parameters. VINs were shown to be capable of computing near optimal paths in 2D mazes and 3D landscapes where the transition model $P(s'|s, a)$ was not provided a priori and had to be learned.

Despite their successes, VINs have several shortcomings. The first is the fact that because the transition model is parameterized by a convolution, it is effectively spatially invariant. This means that the transition model is the same for every grid unit in the map, which does not make sense for many applications. The second shortcoming is that it can require on the order of $k$ iterations to find an optimal path of length $k$. This means that a potentially large number of recursive iterations must be performed to get an estimate of the path to goal. Since the VIN is meant to be thought of as an inner process that the agent performs at every real environment time step, performing large numbers of iterations can cause a large strain on the agents computational demands.

In this paper, we empirically evaluate whether extending VIN can solve its major shortcoming, by (1) untying the weights spatially to create a "Hyper-VIN", where the filter weights of each spatial position are predicted from neighboring cells in the map design, and (2) by increasing the kernel size in order to increase the flow of information spatially and thereby require fewer iterations to find optimal paths. We show that Hyper-VINs, generally on par with the performance of VINs, do worse than architectures which are not structured according to value iteration. We also demonstrate that larger kernel sizes can decrease the stability of VIN, often degrading performance significantly compared to the standard (3,3) kernel used originally.

Additionally, we demonstrate that VIN is often plagued by training instability and random seed sensitivity. Owing to these optimization difficulties, we re-frame VIN as a recurrent-convolutional network, which enables us to replace the unconventional convolution+max-pooling recurrent VIN

update with well-established recurrent operators such as the LSTM update (Hochreiter & Schmidhuber, 1997) . These "LSTMIN" (Long Short-Term Memory Iteration Network) are a more general model that relaxes the inductive bias that forces VINs to perform value-iteration. The LSTMIN is shown to perform at least as well or better than VIN and exhibits less hyperparameter sensitivity and training instability. We also present empirical results in the appendix on 3D VizDoom (Kempka et al., 2016) environments, and a downstream task where we replace the VIN module within a QMDP-net (Karkus et al., 2017) with an LSTMIN.

## 2 METHOD

**Is it necessary to approximate Value-Iteration in the architecture?** One question to ask is whether the inductive biases provided by the VIN are even necessary: is it possible that using alternative, more general architectures might work significantly better than those of the VIN? We can view the VIN within the perspective of a convolutional-recurrent network, updating a recurrent state $(V_{i',j'}^{(t)})$ at every spatial position $(i', j')$ at each iteration:

$$\bar{V}_{i',j'}^{(t)} = \max_a \left( \sum_{i,j} W_{R,i,j}^{\bar{a}} \bar{R}_{i'-i,j'-j} + W_{V,i,j}^{\bar{a}} \bar{V}_{i'-i,j'-j}^{(t-1)} \right) \quad = \omega \left( W_R^{\bar{a}} \bar{R}_{[i',j',3]} + W_V^{\bar{a}} \bar{V}_{[i',j',3]}^{(t-1)} \right)$$

where the notation $X_{[i',j',F]}$ means to take the image patch centered at position $(i', j')$ and kernel size $F$. We can easily replace the recurrent VIN update above with the well-established LSTM update (Hochreiter & Schmidhuber, 1997), whose gated update alleviates many of the problems with standard recurrent networks, but we can still maintain the convolutional properties of the input and recurrent weight matrix:

$$h_{i',j'}^{(t)}, c_{i',j'}^{(t)} = \mathbf{LSTM} \left( W_R^{\bar{a}} \bar{R}_{[i',j',F]} + W_h^{\bar{a}} \bar{h}_{[i',j',F]}^{(t-1)}, c_{i',j'}^{(t-1)} \right)$$

We call the convolutional LSTM (Shi et al., 2015) path planning modules which use this update "LSTM Iteration Networks" (LSTMINs).

**Hyper-VIN:** An issue with the original Value Iteration Network was the use of convolutions to represent the model. This caused the model to effectively be spatially invariant, meaning VINs are incapable of truly solving mazeworld in the same way as value iteration on the true model. The result is that VINs learn a work-around that enables them to deal with non-linearities over the state space: it assigns a huge negative reward to every wall position. This is shown in Figure 7. The large reward gradient between walls and non-walls discourages the model from producing policies that "visit" walls, which would be impossible under the true model. Additionally, the spatial convolution model was fixed and invariant for all mazes, which does not make sense as each MDP in the 2D environments require a different transition kernel based on the maze design.

In this paper we try to alleviate this issue by, first, untying the weights of the spatial convolution and, second, predicting the untied convolution weights directly from the maze design. We call this variant the Hyper-VIN, adopting the naming convention from Hypernetworks, where they also used the mechanism of using a network with weights predicted from another network (Ha et al., 2017). To implement the Hyper-VIN, for each position $(i, j)$ in the environment we predict a convolutional weight matrix from the input map design. The Hyper-VIN update equation then becomes:

$$\bar{V}_{i',j'}^{(t)} = \omega \left( W_R^{\bar{a},i',j'} \bar{R}_{[i',j',3]} + W_V^{\bar{a},i',j'} \bar{V}_{[i',j',3]}^{(t-1)} \right)$$

**Kernel Size** Another issue with the VIN was that Value Iteration potentially requires the number of iterations to be at least as large as the length of the longest path in the environment. This means that VINs might also require significant depth in order to find optimal paths. We additionally tested whether we could reduce the number of iterations by increasing the kernel size $F$ of the various models.

## 3 EXPERIMENTS

We use two metrics to compare the models: %Optimal is the percentage of total states whose predicted paths under the policy estimated by the model has optimal length, and %Success is the per-

centage of total states whose predicted paths under the policy estimated by the model reach the goal state.

**Performance:** The results for VIN, Hyper-VIN, and LSTMIN on 15×15 2D mazes are summarized in Table 1. The best results were from hyperparameters obtained from a sweep over learning rate $\alpha \in \{0.001, 0.005, 0.01\}$, $K \in \{5, 10, 15, 20\}$, and $F \in \{3, 5, 7, 9, 11\}$. In order to make comparison fair between LSTMIN, VIN and Hyper-VIN, we utilized a hidden dimension of 150 for LSTMIN and 600 for VIN and Hyper-VIN, owing to the approximately $4\times$ increase in parameters that LSTMIN contains due to the 4 gates it computes.

For the NEWS mazes, we can see that LSTMIN significantly outperforms VIN and has much lower variance, with most random seeds converging to similar performance values. This suggests that LSTMIN is much easier to optimize, and is less sensitive to initial random seed. It also obtains near perfect performance, showing that LSTMINs are extremely effective path planning modules, despite the update equations not being explicitly designed for such a task. For the Differential Drive mazes, VIN achieves similar near-perfect results as the LSTMIN. Despite the near optimal performance, the VIN models are more difficult to train, as evidenced by the learning curves in Figure 2, where the VIN is consistently slower to converge than the LSTMIN. On the NEWS mazes, Hyper-VIN sometimes fails to recover this optimal solution or even results close to it, despite having value iteration on the true MDP parameters within its hypothesis class. Considering this and the high variance of its results over random seeds suggests that training Hyper-VIN with SGD is significantly challenging.

**Analysis of iteration count and kernel size:** We further evaluated the effect of iteration count $K$ and kernel size $F$ on the variable models. Figure 1 shows %Optimal results of VIN and LSTMIN on 15×15 2D Differential-Drive Mazes for different values of $F$ and $K$. We can see from this figure that optimization of the VIN with large $F$ is significantly more unstable than LSTMIN, and we could observe performance oscillating significantly. Despite this instability, maximum performance is typically near LSTMIN, and since we use early stopping we can recover this performance. Similar results on the Differential Drive mazes are presented in Figure 2. We can see from this figure that increasing kernel size generally improves the performance of LSTMIN, and it achieves near perfect results even with only 5 iterations. On the other hand, increasing the kernel size of VIN does not necessarily help performance, and we see that smaller kernel sizes often have greater performance especially with increasing $K$. VIN with large $F$ also seems to become more unstable and performance oscillates significantly over training epochs.

**Hyper-VIN v.s. VI:** A question that can be asked about Hyper-VIN is if they perform as well (or better) than the actual algorithms they were designed to mimic because the true algorithm is within the model class. This would provide some evidence whether such modules were actually computing the value or whether they acted simply like recurrent networks and computed a less interpretable internal representation.

Table 1 shows results on 15×15 2D mazes, which have relatively straightforward models, only requiring for each position to see if a wall blocks the directly adjacent position in that direction. We can see that Hyper-VIN achieves results often worse than Value Iteration and with large variance over random seeds, demonstrating that the Hyper-VIN is significantly difficult to optimize using SGD.

## 4  CONCLUSION

In this work, we explored some shortcomings of the Value Iteration Network (VIN) as previously designed and tried to extend it to overcome these problems. We re-formulated VIN as a convolutional-recurrent network and replaced the unconventional convolution+max-pooling recurrent update with the well-established LSTM recurrent operator to design an LSTM Iteration Network (LSTMIN). We then presented results showing that the LSTMIN achieves results no worse and often better than VIN and its extensions, and alleviates many of the optimization issues such as sensitivity to random seed and training instability. We have presented experimental results comparing VIN, LSTMIN, Hyper-VIN, and VI on 2D path-planning maze tasks, a 3D navigation task in the video game Doom, and in a downstream task by replacing the VIN planning module in a QMDP-Net (Karkus et al., 2017) with an LSTMIN.

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

## A  EXPERIMENTAL RESULTS

| Model | NEWS | | | | Differential Drive | | | |
| | %Optimal | | %Success | | %Optimal | | %Success | |
| | mean | stdev | mean | stdev | mean | stdev | mean | stdev |
|---|---|---|---|---|---|---|---|---|
| Hyper-VIN | 75.1% | 11.4% | 80.2% | 9.48% | 77.6% | 1.93% | 94.8% | 0.609% |
| VIN | 85.8% | 6.56% | 87.1% | 5.66% | 97.8% | 0.140% | 98.7% | 0.127% |
| LSTMIN | 98.9% | 0.160% | 99.3% | 0.160% | 98.1% | 0.919% | 98.8% | 1.13% |
| VI | 94.2% | - | 94.2% | - | 85.1% | - | 85.1% | - |

Table 1: Performance of each model on $15 \times 15$ 2D mazes, each averaged over 7 different random seeds on the same dataset. These results were attained using iteration count $K = 20$ for all models, filter size $F = 3$ for VIN and Hyper-VIN, and $F = 11$ for LSTMIN.

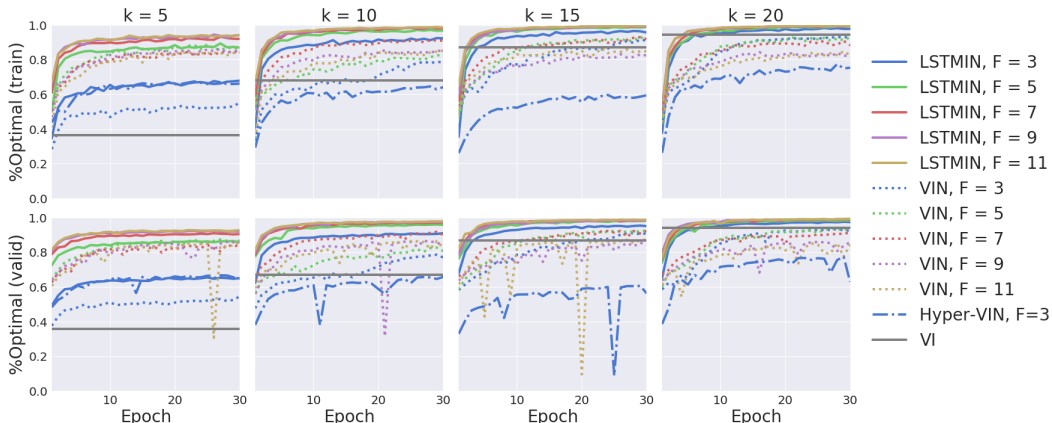

Figure 1: %Optimal results of VIN, Hyper-VIN, VI and LSTMIN on 15×15 2D NEWS Mazes for different values of $F$ and $K$. We can see increasing kernel size generally improves the performance of LSTMIN, and it achieves near perfect results even with only 5 iterations. On the other hand, increasing the kernel size of VIN does not necessarily help performance, and we see that smaller kernel sizes often have greater performance especially with increasing $K$. VIN with large $F$ also seems to become more unstable and performance oscillates significantly. Hyper-VIN was limited to $F = 3$ due to its large memory requirements.

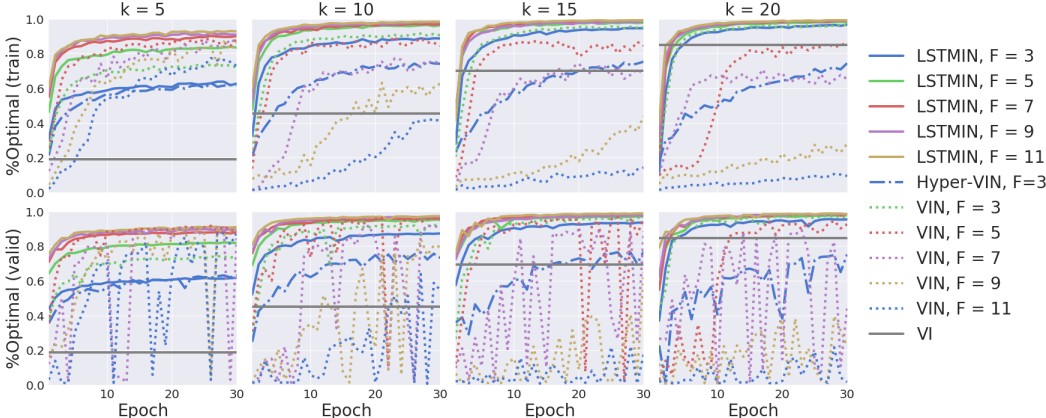

Figure 2: %Optimal results of VIN, Hyper-VIN, VI and LSTMIN on 15×15 2D Differential-Drive Mazes for different values of $F$ and $K$. Optimization of the VIN with large $F$ is significantly more unstable than LSTMIN, and we could observe performance oscillating significantly. Despite this instability, maximum performance is typically near LSTMIN, and since we use early stopping we can recover this performance. Hyper-VIN was limited to $F = 3$ due to its large memory requirements.

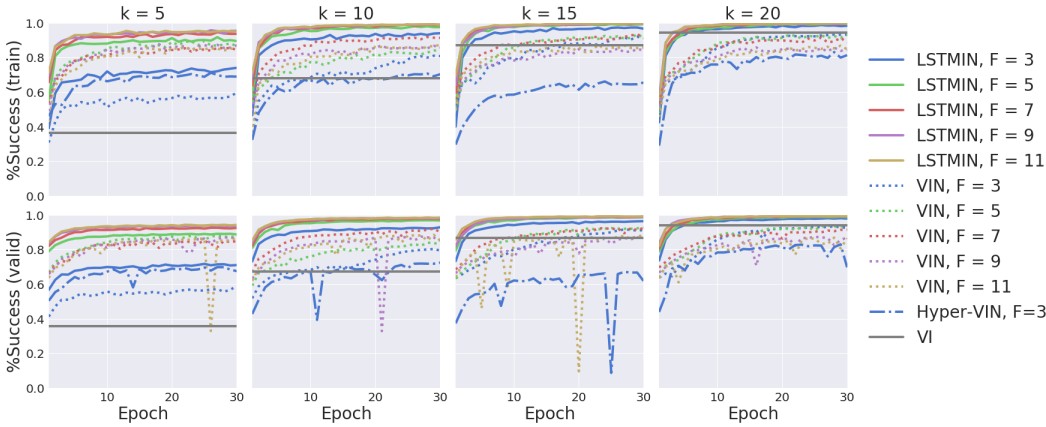

Figure 3: %Success results of VIN, Hyper-VIN, VI and LSTMIN on 15×15 2D NEWS Mazes for different values of $F$ and $K$

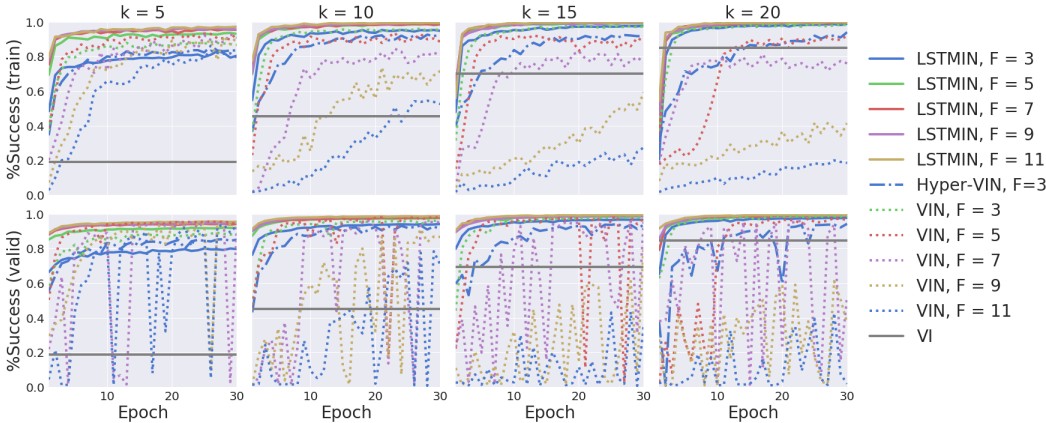

Figure 4: %Success results of VIN, Hyper-VIN, VI and LSTMIN on 15×15 2D Differential-Drive Mazes for different values of $F$ and $K$

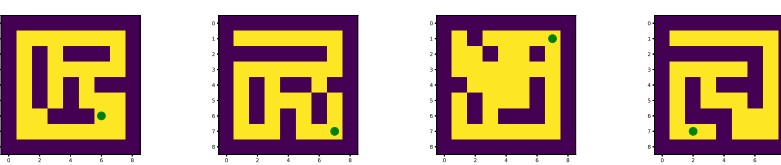

Figure 5: Example mazes demonstrating the 2D maze environment. All mazes are constructed as fully connected trees with a parameter (decimation) that destroys walls with a certain probability. Mazes without any walls destroyed are more difficult because they require longer paths. We can see the third maze from the left has sampled a high decimation, while the far right maze is mostly fully connected with just a single maze wall destroyed. Yellow represents open spaces, purple represents walls and the green circle represents the goal state.

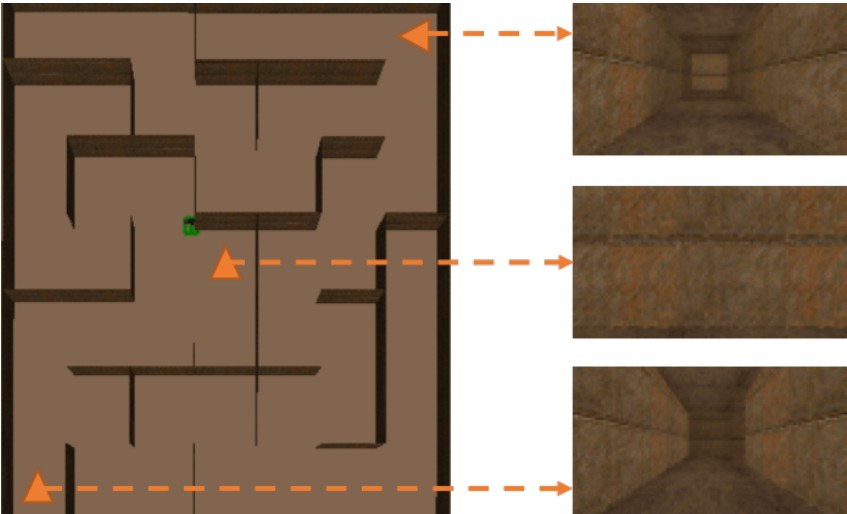

Figure 6: Figure showing a sample 3D Doom maze and examples of screenshots showing the first-person view of the environment at 3 locations.

## B ENVIRONMENTS AND DATASETS

We test our models on a Maze World 2D environment and the VizDoom 3D environment. We used two different maze transition mechanisms: NEWS and Differential Drive. In NEWS, the agent can move North, East, West, or South; in Differential Drive, the agent can move forward along its current orientation, or turn left/right.

The 2D environment is created with a maze generation process that uses Depth-First Search with the Recursive Backtracker algorithm Maze Generation Algorithms (2018) to construct the maze tree. The result is a fully connected maze similar to the one shown in Figure 5. For each maze, we then sample a probability $d$ uniformly from [0,1]. Then, for each wall, with probability $d$ we delete the wall. We sample 25K $15 \times 15$ mazes for training, 5K mazes for validation for evaluating overfitting and 5K mazes for reporting test results.

For our experiments on the 2D maze, the state vector consists of the maze and the goal location, each of which are represented by a binary $15 \times 15$ matrix. We use early stopping based on validation set metrics to choose the final models.

We use the Doom Game Engine and the ViZDoom API (Kempka et al., 2016) to create mazes in a simulated 3D environment. The maze design for the 3D mazes are generated in exactly the same manner as the 2D mazes using Depth-First Search with the Recursive Backtracker algorithm followed by wall pruning with a uniformly sampled probability $d$. The 3D mazes for the 2D designs are then created by using a Python program to generate a Textmap[1] file corresponding to the Doom map. The Textmap file is then used by the Doom Game Engine to create a binary UDMF[2] map.

For each Doom maze, we take RGB screenshots showing the first-person view of the environment at each position and in each orientation. A sample 3D Doom maze and example screenshot images are shown in Figure 6. For a 15x15 maze with 4 orientation, this results in a total of 900 images. These map images are given to the model instead of the 2D map design in the 3D experiments. This setup is similar to the setup used for localization experiments by Chaplot et al. (2018) who argue that these images are easier to obtain as compared to constructing an accurate map design of an environment in the real world. The model needs to learn to infer the map design from these images along with learning to plan, which makes the task very challenging in 3D environments.

---

[1] https://zdoom.org/wiki/TEXTMAP
[2] https://zdoom.org/wiki/Universal_Doom_Map_Format

## C    3D Experiments

| Model | %Optimal | %Success |
|-------|----------|----------|
| VIN | 0.324 | 0.370 |
| LSTMIN | **0.345** | **0.375** |

Table 2: Results for the 3D versions of VIN and LSTMIN models on 3D mazes in the Doom Game Engine.

For the 3D experiments, we use a Convolutional Neural Network (LeCun et al., 1989) for processing the map images. This network consists of two convolutional layers: first layer with 32 filters of size 8x8 and a stride of 4, and second layer with 64 filters of size 4x4 with a stride 2x2, followed by two linear layers of size 512 and 16. This architecture is adapted from previous work which is shown to perform well at playing deathmatches in Doom (Lample & Chaplot, 2017). The 16-dimensional representation for all the 4 orientations at each location is concatenated to create a 64-dimensional representation. These representations of each location are then stacked at the corresponding x-y coordinate to create a map representation of size 64x15x15. The map representation is stacked with the goal map and passed to the VIN or LSTMIN module in the same way as 2D experiments, except that the map design (1x15x15) is replaced by a map representation (64x15x15).

The results in Table 2 shows that LSTMIN performs slightly better than VIN on 3D Mazes. The absolute performance of both the models is low because predicting a 2D top-down transition model from 3D first-person projections of the maze is highly non-linear task and challenging task. Modifications such an auxiliary task for predicting the map design or using depth image instead of RGB image can improve the performance of both VIN and LSTMIN, however those experiments are tangential to the current work.

## D    QMDP-Net Experiments

| Model | Learning Rate | Success Rate |
|-------|---------------|--------------|
| QMDP-net+VIN | 0.00075 | 0.94 |
|  | 0.001 | 0.99 |
|  | 0.0025 | 0.99 |
|  | 0.005 | 0.98 |
| QMDP-net+LSTMIN | 0.00075 | 0.98 |
|  | 0.001 | 1.00 |
|  | 0.0025 | 1.00 |
|  | 0.005 | 1.00 |

Table 3: Results for QMDP-net with either VIN or LSTMIN planner modules on the 10x10 grid-world maze environment. The success rate is averaged over 100 sampled episodes.

In this section, we determine whether we can replace the VIN with an LSTMIN in downstream tasks where a differentiable path-planning module was used. One such application of VIN was as a planning module within a QMDP-net (Karkus et al., 2017). The QMDP-net was used to do belief-state planning in partially observable environments where the current state is not known a-priori. It consisted of a filter module, which implemented a Bayesian filter, and a planning module, which was used to plan ahead in order to choose an optimal action. The planning module consisted of a VIN, and so here we will compare performance between QMDP-nets with VIN and LSTMIN planning modules.

The results used their published code and environment [3] in order to reduce implementation uncertainty and the introduction of bugs, with the only addition of code being a new planner module using LSTMIN. The results are in Table 3 and were evaluated on the grid10 dataset of $10\times10$ grid world.

---

[3]https://github.com/AdaCompNUS/qmdp-net

We evaluated the QMDP-net+VIN and QMDP-net+LSTMIN over several learning rates. From the table, we can see that QMDP-net+LSTMIN achieves consistently higher reward than VIN and is more stable when the learning rates are varied.

# E    VIN REWARD FUNCTION

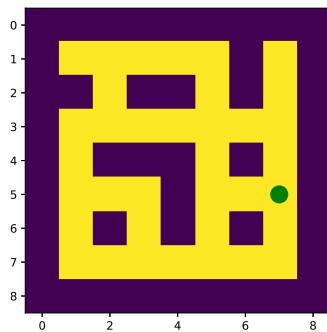 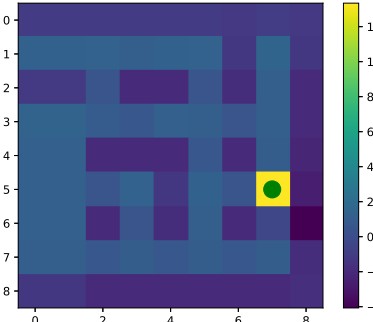

Figure 7: Showing the initial reward vector learned for the 2D maze task on a fully trained VIN. This demonstrates that the VIN gets around the spatial invariance of its model by applying a very large negative reward to states that should never be entered. We can see from the colorbar that there is a difference of around 10 reward units between a wall and an open space. The VIN also assigns very large reward to the goal location (shown with the green circle).

