# OpenReview forum: "LSTM Iteration Networks: An Exploration of Differentiable Path Finding"
_ICLR.cc/2018/Workshop — Accept_

### Official Review · AnonReviewer2 · 2018-03-09
**Value Iteration with LSTM**

**Rating:** 7
**Confidence:** 3

**Review:**

Value Iteration Networks (VINs) for finding optimal policies are hard to train, empirically. They design an architecture based on an LSTM that is more stable during training & robust to random initialization, while obtaining better accuracy. It adds two distinct main components: it replaces the RNN in the standard VIN with an LSTM and it replaces the spatial convolution over the grid with a hyper-VIN that aims to predict these weights in non-convolutional form.

Strengths:
- The hyper-VIN predicting the weights is a very interesting idea
- The motivation for the problems with existing VINs is well done, even in the short space of an abstract
    -- Especially the discussion of how the network gets around the spatial invariance by overweighting the negative reward for walls, and the consequences this has
- Experiments are well-designed to measure the accuracy in the end

Weaknesses:
- RNN -> LSTM is not a significant conceptual leap
- Experimenting with increasing the kernel size is mentioned equally as a contribution with these aspects, but it is also not a conceptual advance
- There are 6 (!) pages of supplemental material and appendices, some of which are quite major and introduce and explain core concepts that are integral to the paper. This does not seem to fit with the page requirement, because the submission would not feel complete without these parts.


Suggested Revisions:
    - General:
        -- Reduce the amount of supplementary material
        -- Emphasis the hyper-VIN aspect, as that is the most novel contribution
       -- The paper needs to be written in a more general, self-contained way.

    - Detailed:
        -- Method section: the first subheading is a complete sentence with a punctuation mark: make it a short phrase

---

### Official Review · AnonReviewer3 · 2018-03-10
**This paper presents LSTMIN as a variant for Value-Iteration-Network, which not only reduces the bias (better learning efficiency) but also controls the variance, as shown mainly by some experiments on 15x15 2D mazes.**

**Rating:** 6
**Confidence:** 3

**Review:**

Value-Iteration-Network (VIN) is a type of deep reinforcement learning implementing Value-Iteration without an explicit model on MDP transition probability, which is successful in finding near optimal paths in 2D mazes and 3D landscapes. A shortcoming of VIN lies in its spatially invariant convolutional kernel of fixed size and the computational cost growth in order with respect to the optimal path lengths.

To address these challenges, by viewing VIN as a recurrent convolutional network, the paper proposed LSTMIN which replaces the iteration map by convolutional LSTM which is suggested by (Shi et al. 2015) for 2D precipitation nowcasting and allows more bias relaxation. In some preliminary experiments on 15x15 2D mazes (News and Differential-Drive), the proposed scheme shows better efficiency in reaching optimal paths with a better variance control at various kernel sizes and depth lengths than the VIN and its naive extension Hyper-VIN by heterogeneous weight relaxations. In appendix further empirical results on 3D mazes and with QMDP-Nets also show that LSTMIN performs slightly better than VIN.

In a summary, this is an interesting exploration on LSTM-based value iteration scheme.  Some questions include:

1. As both VIN and LSTMIN mimic the VI algorithm, could you briefly discuss why the original VI algorithm does not effectively find the optimal paths in the 2D mazes while these deep networks versions can have supreme performance than the original VI?

2. There are other recurrent structures like GRU, are they performing similarly as LSTM used in this work?

---

### Official Review · AnonReviewer1 · 2018-03-11
**interesting paper**

**Rating:** 7
**Confidence:** 3

**Review:**

From the content point of view, a nice study and an interesting comparison. One issue may be the notation used, there is nothing wrong with it but it is only half-heartedly explained. I would have preferred a systematic and more complete description of it.

I am not sure if it was intended that extended abstracts on ICLR have an abstract -- personally I don't think a 3 page extended abstract should have one. The paper kind of works around the page limit also by moving results into the appendix - removing the abstract may introduce space for some more work (or to put the results back in). Having said that, the information in the appendix is quite valuable, and the paper is interesting to read and a good study for a workshop contribution. For a full paper as a follow-up to this one I would like to see more comparisons and evaluations, eg other planning approaches, different environments.

---

### Decision · Program_Chairs · 2018-03-20
**ICLR 2018 Workshop Acceptance Decision**

**Decision:**

Accept

**Comment:**

Congratulations, your paper was accepted to the ICLR workshop.